# Estimating Body Weight in Captive Rabbits Based on Improved Mask RCNN

**Enze Duan [1], Hongyun Hao [2], Shida Zhao [1], Hongying Wang [2] and Zongchun Bai [1,]***

[1]  Agricultural Facilities and Equipment Research Institute, Jiangsu Academy of Agriculture Science, Nanjing 210014, China
[2]  College of Engineering, China Agriculture University, Beijing 100082, China
*    Correspondence: vipmaple@126.com

**Abstract:** Automated body weight (BW) estimation is an important indicator to reflect the automation level of breeding, which can effectively reduce the damage to animals in the breeding process. In order to manage meat rabbits accurately, reduce the frequency of manual intervention, and improve the intelligent of meat rabbit breeding, this study constructed a meat rabbit weight estimation system to replace manual weighing. The system consists of a meat rabbit image acquisition robot and a weight estimation model. The robot stops at each cage in turn and takes a top view of the rabbit through an RGB camera. The images from the robot are automatically processed in the weight estimation model, which consists of the meat rabbit segmentation network based on improved Mask RCNN and the BW fitting network. Attention mechanism, PointRend algorithm, and improved activation function are proposed to improve the performance of Mask RCNN. Six morphological parameters (relative projected area, contour perimeter, body length, body width, skeleton length, and curvature) are extracted from the obtained mask, and are sent into the BW fitting network based on SVR-SSA-BPNN. The experiment shows that the system achieves a 4.3% relative error and 172.7 g average absolute error in BW estimation for 441 rabbits, while the meat rabbit segmentation network achieves a 99.1% mean average precision (mAP) and a 98.7% mean pixel accuracy (MPA). The system provides technical support for automatic BW estimation of meat rabbits in commercial breeding, which is helpful to promote precision breeding.

**Keywords:** deep learning; Mask RCNN; breeding robot; weight estimation; rabbit



## 1. Introduction

China has always been committed to improving the diversity and health of consumable meat to meet people's living needs, which places high demands on the standardization and automation of breeding. Animal weight is an important parameter in breeding management [1,2], which is of great significance in evaluating breeding efficiency and making breeding decisions. In the rabbit cage breeding industry, equipment and scale reduce the feasibility of manually weighing animals. With the progress of machine vision and artificial neural network technology, it is the development direction of the industry to use deep learning technology and sensor technology to estimate the weight of meat rabbits.

At present, there has been some exploration of the application of machine vision technology in animal BW estimation. Using two-dimensional images to fit animal weight is the most classic method. Amraei et al. [3] use an ellipse fitting algorithm and Chan–Vese method to obtain chicken body contour, and six physical extracted features were used to fit the BW by different ANN techniques. However, the features mentioned in this research were not extracted from the chickens directly, which reduced the fitting accuracy. Kashiha et al. [4] constructed a fitting model based on the SISOTF model to calculate regression of the pig BW with the fitted ellipse area by calculating the fitted ellipse area on the back of pigs, and the results showed that the fitting accuracy of the model was

97.2%. Zhuang et al. [5] used YOLOv3 and FCN to locate and segment the broilers depth images and fitted five body size features to estimate the BW of broilers. The absolute error was 0.01–0.32 kg, but this study was also based on the laboratory environment. Three-dimensional cameras are currently a hot technology for estimating body weight based on visual technology. Wang et al. [6] obtained the broilers' depth image by depth camera, and extracted the projection area, perimeter, eccentricity, back width, and volume by computer vision. The broiler BW was predicted by BP neural network, and the optimal fitting degree was 0.994. However, the limitation of this study is that this research is based on laboratory environment. Kuzuhara et al. [7] used a 3D camera to obtain the back posture of dairy cows to predict six indexes, body condition score, BW, milk yield, milk fat, and milk protein, and the correlation between body size characteristics and indexes was analyzed by principal component analysis. However, due to the high requirements for the working environment and imaging distance of 3D cameras, it is difficult to apply the researches in large-scale commercial aquaculture. Overall, existing researches show that it is feasible to use machine vision to estimate animal BW, but most of them are not based on commercial production environments, and their image datasets depend on manual work. In addition, the accuracy of image segmentation and the selection of animal image features have a significant impact on animal weight estimation.

Rabbit BWs are critical physical growth attributes to assess production efficacy, which can help to eliminate individuals with low production performance in time. At present, the manual weighing of meat rabbits in breeding farms consumes a significant amount of resources and can further exacerbate rabbit stress, ultimately leading to illnesses and reduced productivity. Automated BW estimation can not only reduce the loss of farm resources, but also help to grasp the health status of individual animals and provide an accurate big data basis for breeding management. We have already studied a meat rabbit BW estimation model based on deep learning in 2021. The model has achieved good accuracy in the dataset [8], but still has the following shortcomings: (1) the images of the dataset were acquired manually rather than automatically; (2) the experimental environment was a manual turnover box rather than a cage used in actual breeding; (3) the accuracy of the model needs to be improved.

In order to solve the problems mentioned in the above researches, we conducted another study on the meat rabbit BW estimation in a commercial breeding environment. This study has two main contributions. Firstly, a self-inspection robot was designed for meat rabbit farming facilities, enhancing image collection convenience and intelligence while reducing human impact on animals. This robot technology has a potential application in other animal farming scenarios. Secondly, a machine vision-based model was developed to estimate meat rabbit weight accurately. The instance segmentation accuracy was improved by improved Mask RCNN, and the effect of different network optimization modules on performance when segmenting rabbit targets was discussed. Additionally, six image features of meat rabbit instances were proposed and correlated with meat rabbit weight. An improved fitting network was constructed, leading to enhanced fitting accuracy compared to the original network. This study is the first to apply robot technology and deep learning technology in the meat rabbit breeding field, achieving automatic and non-contact acquisition of meat rabbit weight.

## 2. Materials and Methods

### 2.1. Data Acquisition

The images were captured in a meat rabbit breeding farm in Henan Province, China, over several days in October 2021. An autonomous inspection robot based on ROS [9] was designed to collect images in rabbit cages. Magnetic navigation was used for the guidance of the robot to travel and stop along the designated route. The 3D structure and operation status of the robot are shown in Figure 1.

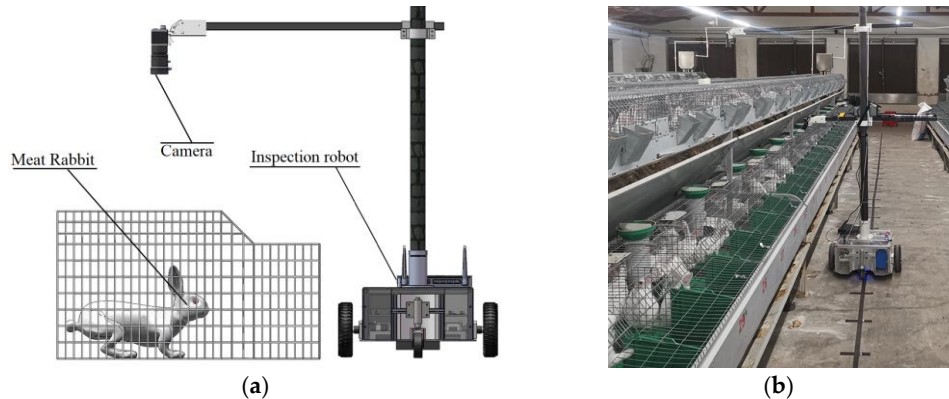

**Figure 1.** The autonomous inspection robot. (**a**) 3D structure. (**b**) Operation status.

The robotic system employs a master-slave control architecture. The main controller of the robot is a mini PC (Dell OptiPlex 7090MFF, Dell Inc., Xiamen, China), which deploys Ubuntu 20.04 and Windows 10 virtual environment to control the movement, parking, image capture, and save of the robot. The slave controller is an embedded controller (STM32, STMicroelectronics, Shanghai, China) which receives the motor speed and steering information sent by the main controller and controls the wheel motor. The control system is divided into main program, image capture program, sensor control program, and PID control program, which are written in Python and C++. The connection and communication relationship of hardware are shown in Figure 2.

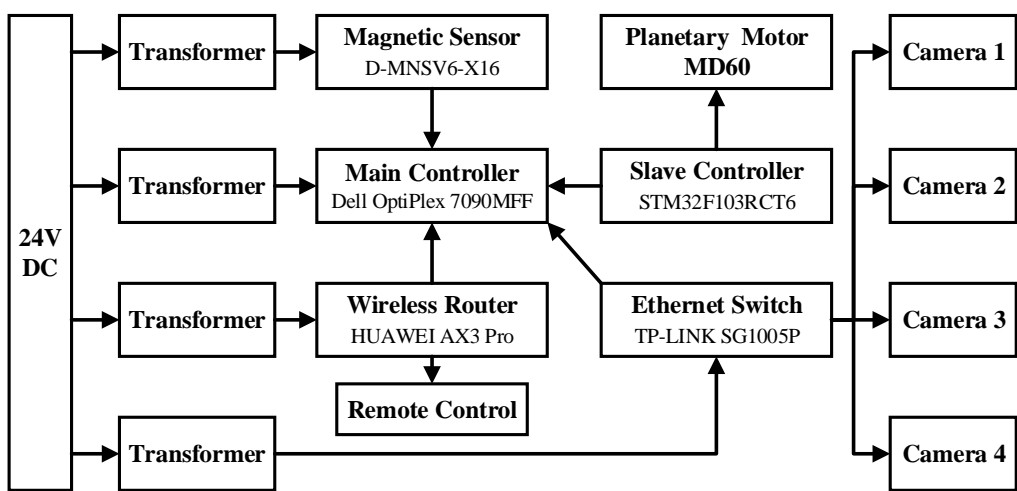

**Figure 2.** The connection and communication relationship of hardware.

Figure 3 illustrates how ROS is utilized in robot control. Each component of the robot communicates, computes, and controls information through ROS. The robot_control node served as the central control hub that publishes data through the nodelet_manager node. The velocity_smooth node is responsible for smoothing the wheel speed; the camera_capture node controls all cameras; and the platform_control node is another major control node responsible for managing the magnetic strip sensor signal (meg_sensor), and controlling vehicle speed and steering (robot_moving).

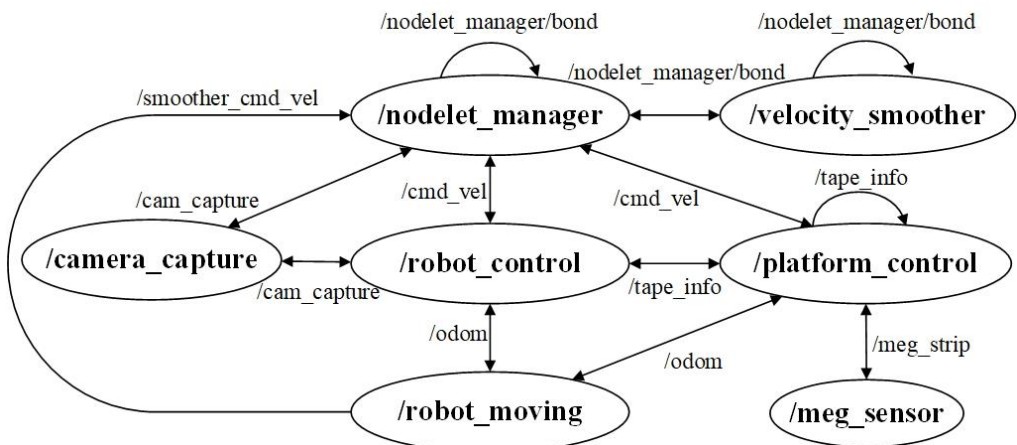

**Figure 3.** Computation graph of the ROS.

The patrol logic of the robot is based on 16 magnetic induction lamps, where each induction lamp represents an analog quantity from 1 to 16. During operation, the magnetic sensor continuously returns analog signals from 3 induction lamps. The average feedback value of these three induction lamps is 8.5 when the inspection robot is located directly above the magnetic strip track. If the inspection robot deviates from the magnetic strip track, the difference between the average analog value and 8.5 will be used for the proportional-integral-derivative (PID) controller. The main controller calculates the wheel speed and rotation direction based on the PID result. When more than 10 induction lamps detect magnetic signals, the robot has reached the sampling point, and it will park while the camera captures images.

Using the classical engineering tuning method, the PID parameters were determined as follows: $K_p = 0.0175$, $K_i = 0$, and $K_d = 0$. The robot's trajectory on the 40 m magnetic strip track is shown in Figure 4, which indicates a maximum deviation of 2.43 cm. The driving stability of the robot meets the requirements for image acquisition.

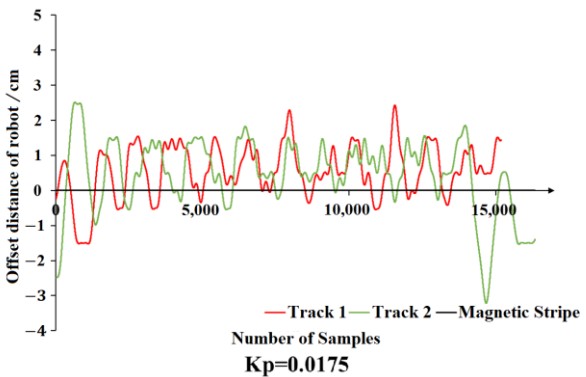

**Figure 4.** The traveling tracks of patrol robot on 40 m track.

The inspection robot was used to capture images in this research. We focus on capture images of the upper cages in the breeding cages, because the upper cages breed pregnant rabbits, growing rabbits, and reserve rabbits. Those rabbits are not only easier to be captured, but also their BW is more significant to the breeding of meat rabbits. For the other rabbits, such as baby rabbits and lactating rabbits, their BW fluctuates widely in a short time and was therefore less statistically significant. The industrial camera model used in this study is SONY CG240C, the exposure time is 23 ms, and the highlight is 20 dB.

### 2.2. Meat Rabbit Instance Segmentation Network Based on Improved Mask RCNN

2.2.1. The Structure of Mask RCNN

Mask RCNN [10] was a classic image instance segmentation network which could extract region and classify bounding box, respectively, through a two-stage network. Mask RCNN could effectively perform object classification, semantic classification, and instance segmentation. The structure of Mask RCNN includes backbone network, Region Proposal Network (RPN) [11], RoIAlign, and functional network. The structure of Mask RCNN used in this study is shown in Figure 5.

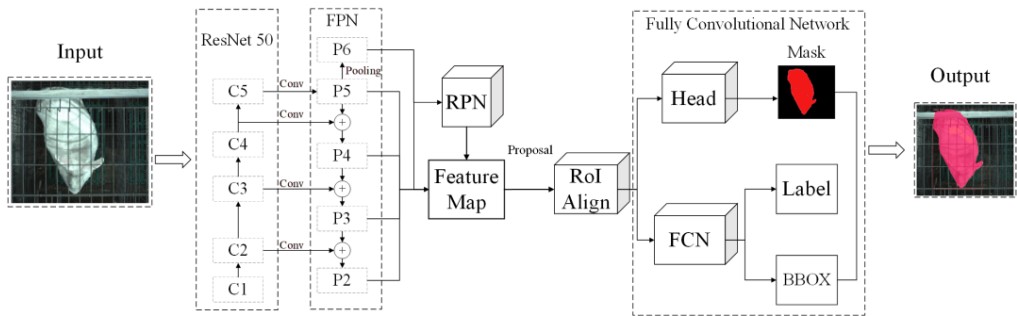

**Figure 5.** The structure of rabbit segmentation network based on Mask RCNN.

The backbone network convolves the input images and outputs the feature maps of different dimensions and sizes to extract image features in advance. Deep residual network [12] and Feature Pyramid Network (FPN) make up the backbone network. Since the segmented objects in this study were meat rabbits, FPN based on ResNet-50 was chosen as the backbone network to speed up the model operation. ResNet-50 was composed of two residual modules ConvBlock and IdentityBlock, and through learning the residual between input and output of different network layers, the problem of gradient disappearance had been solved.

The Region Proposal Network used a sliding window to scan the feature maps from the backbone network, and randomly generated plenty of anchors with random sizes and random positions on the feature maps. The generated anchors would be regressed with the manual labeling images to calculate the correction value and filtered out of the regions of interest (RoI) areas where the rabbits exist.

RoIAlign was used to map the pixels before and after scaling with bilinear interpolation, which significantly improved the detection of small objects.

While Mask RCNN had been proven to have a great performance on instance segmentation, it did not perform well on the segment of object edges. Convolution and pooling during backpropagation caused the loss of spatial location information of key pixels, and the spatial relationships between different size feature maps were not effectively used during segmentation, which mean the segmentation accuracy of small objects was higher than that of large objects.

2.2.2. Optimized Backbone Network Based on Attention Mechanism

In order to reduce the computing power waste caused by using the same weight in each part of the input image during the backbone network extracted feature maps, this study introduced an attention mechanism [13] to optimize the backbone network. The attention mechanism consists of channel attention module (CAM) and spatial attention module (SAM). During the convolution process, the attention mechanism assigns different computational weights to each part of the input image. The structural relationship between attention module and ConvBlock was shown in Figure 6.

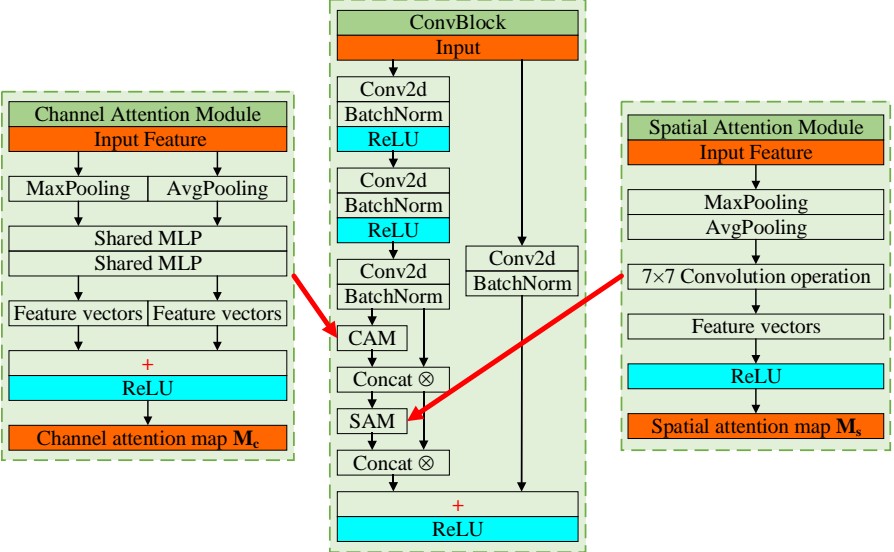

**Figure 6.** The structure of residual network with attention module.

CAM takes the feature F which was obtained by ConvBlock convolute three times as the input, and F was used to extract the channel description through max-pooling and average-pooling, respectively. The corresponding feature vectors were obtained by the shared multi-layer perceptron (MLP). After addition and activation, the final vector was multiplied with F as the input of SAM.

SAM utilizes the outputs of maximum pooling and average pooling that are pooled along the channels axis, and forward them to a convolution layer and output feature vectors. The ReLU [14] function was used to activate the final vectors.

This study adds the attention mechanism to the backbone network, which can help to optimize the computing power distribution in convolution of ResNet-50, and improves the training speed and performance of the network.

### 2.2.3. Optimized Backbone Network Based on Improved Activation Function

The activation function maps the input of the neuron to the output during convolution, and converts the linear operation of the convolution to the nonlinear output, which can enhance the updating ability of the network weight. A suitable activation function should have the following characteristics: (1) Prevent gradient dispersion. The derivative function of the activation function should not be 0 on the positive and negative axes of the x-axis to make sure that the network weight can be updated; (2) Sparsity. The derivative function of the activation function should be close to 0 on the negative axis of the x-axis to ensure that the updating speed of the network weight is not too fast.

Mask RCNN uses ReLU as the activation function. Since ReLU is 0 on the negative axis of the x-axis, the network weight will not be updated when the convolution result is negative. Therefore, this study introduced Mish function [15] as the activation function of ResNet-50. The definitions of ReLU and Mish function are shown as follows:

$$\text{ReLU: } f(x) = \begin{cases} 0 & x < 0 \\ x & x \geq 0 \end{cases} \tag{1}$$

$$\text{Mish: } f(x) = x\tanh(\ln(1 + e^x)) \tag{2}$$

The function image of ReLU and Mish is shown in Figure 7:

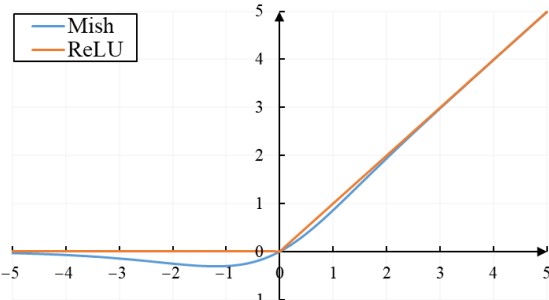

**Figure 7.** The function image of ReLU and Mish.

Figure 7 shows that the characteristic of the Mish function is similar to that of the ReLU function on the positive axis of the x-axis; that is, the derivative function is close to 1. On the negative axis of the x-axis, the Mish function has a lower limit, which makes the network have regularization ability. Therefore, this study chose the Mish function instead of the ReLU function as the activation function of the network, which can improve network accuracy during iteration.

2.2.4. Optimized Boundary Feature Based on PointRend Technique

In the process of classifying mask pixels, the images were divided into a large number of grids, and the category of each grid was predicted by the network. However, during this process, the network allocated the same computing power to each grid, which caused the waste of computing power. For grids located inside the mask, it is easy to oversample, and for these grids located at the mask boundary which need to be focused on, the sampling depth of the network is not enough; this increases the loss of segmentation on the mask boundary.

This study introduced PointRend [16] algorithm to optimize the process of mask pixels classification. A subdivision strategy was used to select a non-uniform set of points at which to compute pixels' labels, which concentrate the calculation to the mask boundary. The modules in the PointRend consist of point selection, point-wise feature extraction, and point head.

The point selection module aims to achieve flexible and adaptive point selection for predicting segmentation labels, with a focus on selecting points that are near high-frequency ports, such as object boundaries. During inference, a coarse-to-fine rendering strategy is employed, where the points on a regular grid are first predicted with the coarsest resolution. The low spatial resolution feature map is then up-sampled using bilinear interpolation to achieve the desired resolution. Subsequently, on the denser grid, the most uncertain points whose degree of confidence is less than 0.5 (e.g., confidence interval) are selected. The points are selected as follows:

$$n_i^* = \underset{n_i}{\mathrm{argmin}} |p(n_i) - 0.5| \tag{3}$$

where $n_i^*$ is the selected point; and $p(n_i)$ is the probability for point $n_i$ belonging to the mask boundary.

The point-wise feature extraction module is used to extract the feature of $\beta N$ points as the input of the next stage. The features of each point were composed of fine-grained features and coarse prediction features. The fine-grained features contained the segmentation details, and can be computed by bilinear interpolation on the feature map. The coarse predicted feature is computed by bilinear interpolation on the feature map, and contains semantic information. The features of points will be sent to the point head module.

The point head module is a Multilayer Perceptron (MLP) with 3 hidden layers and 256 channels. By iteratively convolving the point-wise features of the different size feature maps, the point head module connects the fine-grained features and coarse prediction

features, up-sampling the resolution of the feature map to the target resolution. Finally, the boundaries of the rabbit's mask are segmented finely.

### 2.2.5. Data Acquisition and Dataset Construction

The rabbit images were captured by the autonomous inspection platform from actual production. All the rabbits whose images were captured were also weighed, and the dataset was constructed. The age of meat rabbits used in this study ranged from 34 days to 1180 days, which improved the effectiveness of the model for meat rabbits at different growth stages.

Images were clipped to 1024 pixels × 1024 pixels and enhanced by the RetinexNet [17] algorithm. In this study, the shooting angles of all images were fixed, so the images were only mirrored and Gaussian noise [18] was added to expand the dataset, in order to prevent the accuracy of the model from being affected by the types of training images. Labelme was used to label 900 images, 80% (720 images) of them were selected as the training dataset, and the remaining 20% (180 images) were used as the verification dataset.

### 2.3. Feature Extraction of Rabbit Body

#### 2.3.1. Pretreatment of Rabbit Mask Image

After the captured images were processed by the meat rabbit instance segmentation network, most of the obtained mask images contained one complete rabbit mask. However, in a small number of mask images, there may have holes in the rabbit mask, or multiple rabbits were segmented, which are shown in Figure 8, and these mask images need to be repaired.

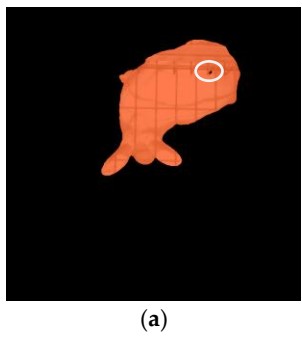 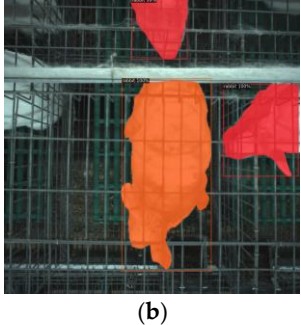

(**a**)          (**b**)

**Figure 8.** Segment results which need to be repaired. (**a**) Mask with hole (mark in white circle). (**b**) Mask image with multiple rabbit mask.

The repair method is as follows. The images were binarized first. OpenCV was used to calculate the pixel number of mask contours in the image, and the number of pixels in each contour was counted. Due to the camera being directly above the cage, the mask with the largest number of contour pixels was the target rabbit number, and the other masks were removed. For the holes in the mask, the contours of the target rabbit mask were all zero, and the images were repaired.

The areas of rabbit ears and legs in the image were large, but the actual weight was small, so these parts need to be removed from the mask. The continuous morphological opening based on kernel with adaptive size was used to remove the rabbit ears and legs. For each rabbit, the size of the kernel which was used in the morphological opening was different. Proven by experiment, the original kernel size was taken as 1/1000 of the area of the rabbit mask in the first morphological opening, which can make the morphological opening effective in different sizes of meat rabbits. To smooth the contour of rabbit masks, the morphological opening is operated twice on the mask; the size of the kernel was half that of the previous one.

### 2.3.2. Feature Extraction

Six rabbit body features including relative projected area, contour perimeter, body length, body width, skeleton length, and curvature were extracted from the preprocessed mask images. The extraction methods are shown as follows:

1. Relative projected area can be represented by the number of non-zero pixels in the mask contour, which can be recorded as $S_R$;

2. Contour perimeter can be represented by the number of pixels of the rabbit mask contour, which can be recorded as $L_C$;

3. Body length can be represented by the length of the long side of the minimum bounding rectangle of the rabbit mask contour, which can be recorded as $L_R$;

4. Body width can be represented by the length of the short side of the minimum bounding rectangle of the rabbit mask contour, which can be recorded as $L_W$;

5. Skeleton length can be represented by the pixel numbers of the rabbit mask skeleton; the skeleton was calculated by the thinning algorithm [19]. For these skeletons with branches, a continuous morphological opening was used to the masks to smooth the contour, until there was no branch on the skeletons. The skeleton length is recorded as $L_B$;

6. Curvature represented the shape bends or twists of rabbits. Graham scanning method was used to calculate the convex hull of the skeleton, and the area of the convex hull reflects the bend of the skeleton. In order to reduce the influence of rabbit body size, curvature is recorded as $C$ which can be calculated as the ratio of convex hull area to skeleton length.

### 2.4. Rabbit Weight Regression Model Based on Optimized Machine Learning

SVR (Support Vector Regression) [20] and BPNN (Back Propagation Neural Net) [21] were widely used in data regression. SVR has strong robustness and boundary adaptability, but it also has high requirements for parameters and data. BPNN can operate complex nonlinear problems, and has low requirements for the direct correlation of data, but it is easy to fall into local minimum, and it has high requirements for data volume. To integrate the advantages of the two models, the prediction results of SVR and BPNN [22] were weighted by BPNN. SSA (Sparrow search algorithm) [23] was used to optimize the initial weights of BPNN.

### 2.4.1. Sparrow Search Algorithm

SSA is a swarm optimization approach proposed, which is inspired by the group wisdom, foraging, and anti-predation behaviors of sparrows. The data population were divided into producer, scrounger, and vigilante: the producers were responsible for global search of the network; the scrounger were responsible for local search of the network; and vigilante were responsible for controlling the population movement, keeping their position as close as possible to the center of the population. The position update formulas for these individuals were described as follows:

For producers:

$$X_{i,j}^{t+1} = \begin{cases} X_{i,j}^t \cdot \exp(-\frac{i}{\alpha \cdot iter_{\max}}) & R_2 < ST \\ X_{i,j}^t + Q \cdot L & R_2 \geq ST \end{cases} \tag{4}$$

For scroungers:

$$X_{i,j}^{t+1} = \begin{cases} Q \cdot \exp(\frac{X_{worst}^t - X_{i,j}^t}{i^2}) & if \ \ i > \frac{n}{2} \\ X_P^{t+1} + \left| X_{i,j}^t - X_P^{t+1} \right| \cdot A^+ \cdot L & otherwise \end{cases} \tag{5}$$

For vigilante:

$$X_{i,j}^{t+1} = \begin{cases} X_{best}^t + \beta \cdot \left| X_{i,j}^t - X_{best}^t \right| & if \quad f_i > f_g \\ X_{i,j}^t + K \cdot \left( \dfrac{\left| X_{i,j}^t - X_{worst}^t \right|}{(f_i - f_w) + \varepsilon} \right) & if \quad f_i = f_g \end{cases} \qquad (6)$$

where $t$ indicates the current iteration; $j$ is the data dimension; $iter_{max}$ is a constant with the largest number of iterations; $X_{ij}^t$ represents the value of the $j$-th dimension of the $i$-th sparrow at iteration $t$. $\alpha \in (0, 1]$ is a random number; $R_2$ ($R_2 \in [0, 1]$) and $ST$ ($ST \in [0.5, 1.0]$) represent the alarm value and the safety threshold, respectively; $Q$ is a random number which obeys normal distribution. $L$ shows a matrix of $1 \times d$ for which each element inside is 1; $X_{worst}$ denotes the current global worst location; $X_p$ is the optimal position occupied by the producer; $A$ represents a matrix of $1 \times d$ for which each element inside is randomly assigned 1 or $-1$, and $A^+ = A^T (AA^T)^{-1}$; $n$ is population size; $X_{best}$ is the current global optimal location; $\beta$ is a random number that controls step size parameter; $f_i$ is the fitness value of the present sparrow. $f_g$ and $f_w$ are the current global best and worst fitness values, respectively; $K \in [-1, 1]$ is a random number; $\varepsilon$ is the smallest constant so as to avoid zero-division-error.

By iteratively calculating the optimal fitness value of the population, SSA was used to optimize BPNN. The accuracy of BPNN was improved by using the SSA output as the initial weight and threshold. The training process of SSA-BPNN is shown in Figure 9.

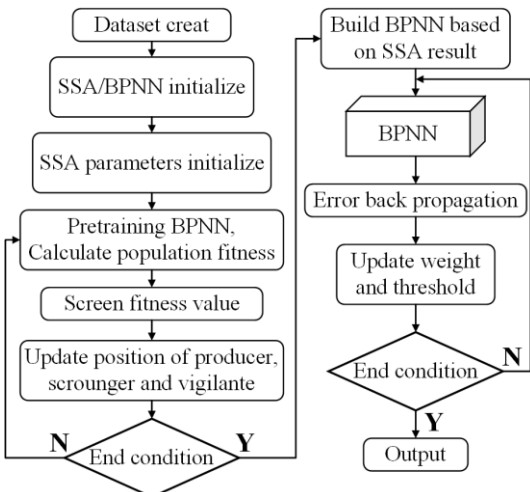

**Figure 9.** The training process of SSA-BPNN.

2.4.2. Meat Rabbit Weight Regression Model Based on SVR-SSA-BPNN

The SVR-SSA-BPNN model proposed in this study improved the accuracy of regression through weight optimization and algorithm weighting. The method is shown as follows:

1. Map the data to the (0, 1) with the Mapminmax standardization method, and divide the original dataset into a training set T and a test set V in an 8:2 ratio;

2. Divide the training set T into training set $T_1$ and test set $V_1$ in a 6:4 ratio;

3. Train SVR and SSA-BPNN with training set $T_1$, and record the obtained model as $M_S$ and $M_B$, respectively;

4. Use model $M_S$ and $M_B$ to predict dataset $V_1$, and record the obtained result as $R_S$ and $R_B$;

5. Train new SSA-BPNN with $R_S$ and $R_B$ as input and $R_{V1}$ as output, where the obtained model $M_B'$ is the weighted result of SVR and SSA-BPNN;

6. Use dataset V to validate the model $M_S$-$M_B$-$M_B'$ and evaluate its performance.

The training process of SVR-SSA-BPNN model is shown in Figure 10.

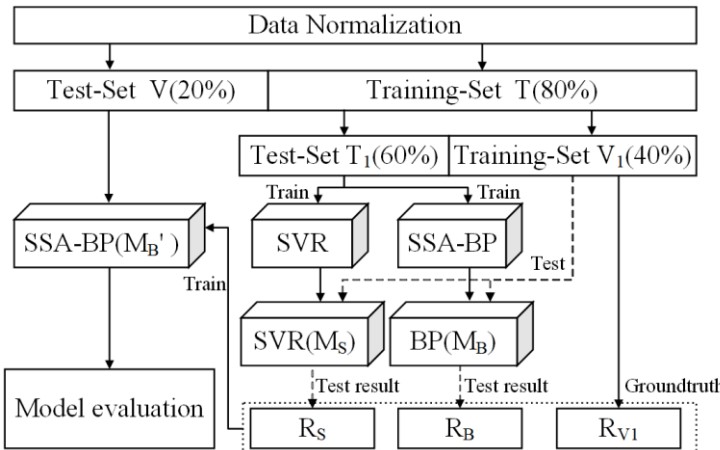

**Figure 10.** The training process of SVR-SSA-BPNN.

*2.5. Algorithm Platform*

The model was trained on a computer with Inter(R) Core(TM) i7-9700 CPU and NVIDIA GeForce RTX2080 GPU; the system is Ubuntu 20.04. The instance segmentation algorithms were implemented based on the PyTorch deep learning framework, and the weight prediction algorithm was implemented with MATLAB R2016a.

## 3. Results

*3.1. Performance of Meat Rabbit Instance Segmentation Network*

3.1.1. Network Group and Performance Evaluation Indexes

To compare the performance of optimized networks with various structures and to identify the most suitable network, the meat rabbit instance segmentation network was classified into the following four groups: (1) Unoptimized Mask RCNN, noted as A; (2) Mask RCNN improved with attention mechanism and Mish, noted as B; (3) Mask RCNN improved with attention mechanism and PointRend, noted as C; (4) Mask RCNN improved with attention mechanism, Mish, and PointRend, noted as D.

The evaluation index system includes AP (average precision), MPA (mean pixel accuracy), and the time used for single image inference. AP describes the classification accuracy of the model for rabbit objects; MPA describes the classification accuracy of the model for rabbit mask pixels; and the time used for single image inference describes the operation speed of the model. The indexes were shown as follows:

$$AP = \frac{TP}{TP + FP} \tag{7}$$

where *TP* is true positive; *FP* is false positive.

$$MPA = \frac{1}{k}\sum_{i=0}^{k}\frac{p_{ii}}{\sum_{j=0}^{k}p_{ij}} \tag{8}$$

Here, $k$ is the number of labels including background; $p_{ii}$ is the total number of pixels whose real pixel class is $i$ and predicted as $i$; and $p_{ij}$ is the total number of the pixel whose real pixel class is $i$ but predicted as $j$.

3.1.2. Performance of Improved Mask RCNN

A validation dataset was used to train four meat rabbit instance segmentation networks, and the results were shown in Table 1.

**Table 1.** Performance of four meat rabbit instance segmentation networks.

| Model | AP/% | MPA/% | The Time Used for Single Image/s $\cdot$ f$^{-1}$ |
|---|---|---|---|
| A | 99.0 | 94.8 | 0.12 |
| B | 99.3 | 95.1 | 0.15 |
| C | 99.1 | 98.7 | 0.13 |
| D | 99.2 | 95.7 | 0.17 |

The results in Table 1 show that the AP values of four models were all higher than 99% when IoU (Intersection over Union) was 0.85, and the AP value of model B was the highest, reaching 99.3%, but there was not much difference among other networks. The AP value results show that all the models have a high accuracy in rabbit recognition, mainly because the distinction of rabbits in the cage is high and image preprocessing further highlights the differences between foreground and background.

The difference in MPA values of different networks is significant. The MPA value of model A is the lowest, 94.8%, which indicates that the three optimization methods have improved the segmentation accuracy. Model C has the highest MPA value, 98.7%. The differences in MPA values of models BCD show that the three optimization methods have improved the performance of the Mask RCNN; however, the MPA value of model D has decreased on the basis of structure C. This potentially arises due to the fact that the modification of the activation function enhanced the performance of the backbone, but the mask branch suffered a decrease in obtaining high-resolution feature maps during back-propagation.

In terms of single image inference time, the lowest average prediction speed is 0.12 s/f, while the highest average prediction speed is 0.17 s/f, which indicates a slight increase in computation for the optimized network. When comparing the various models, it is evident that the addition of the Mish function has increased the inference speed of the model by more than 0.03 s. This is because the calculation of the Mish function is greater than that of the ReLU when it increases nonlinearity of the output from the network layers. The comparison between models' AC shows that PointRend and the attention mechanism have increased the average prediction speed, but not significantly.

The training loss was used to reflect the training performance of the network as the number of iterations increases, which is shown in Figure 11.

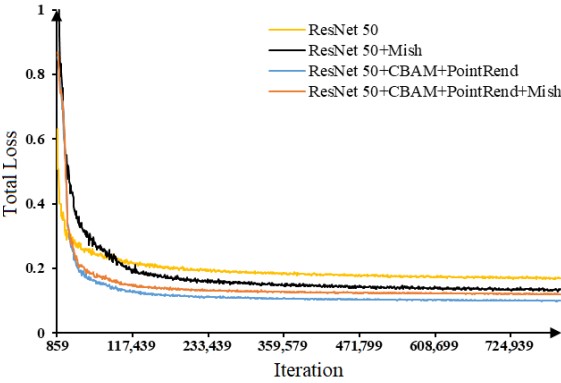

**Figure 11.** The training loss of four models.

The training loss shows that model C had the lowest loss value, converging to 0.101, which is consistent with the conclusion in Table 1.

Four models were used to infer a randomly selected rabbit image from the test set, and the results are shown in Figure 12.

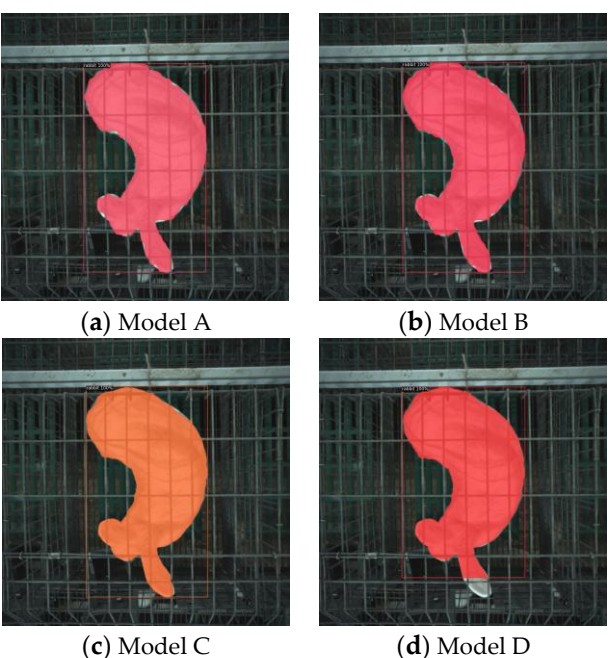

**Figure 12.** Segmentation results of four models.

Figure 12 demonstrates that the segmentation accuracy at the edges of the mask was better for model CD than for models AB, which indicates that PointRend and the attention mechanism can improve the performance of contour edge segmentation. Furthermore, the comparison of model CD's segmentation results showed that model D had a gap in the mask at the rabbit's ears, which was related to the regression anchor, implying that the Mish function might interfere with the accuracy of the feature pyramid network while selecting the regression anchors, where the smaller anchor was easily judged as the overall rabbit regression anchor. This also indicates that model C has the best segmentation performance.

Therefore, this study concluded that Mask RCNN with attention mechanism and PointRend had the best performance of instance segmentation. Additionally, subsequent research in this study was based on the results of this network.

### 3.2. Performance of Meat Rabbit Weight Estimation Network

3.2.1. The Dataset of Meat Rabbit Weight Estimation Network

The obtained rabbit images were processed by the meat rabbit instance segmentation network, and the method described in Section 2.3 was used to extract six body features from the rabbit masks. Considering the correlation between $D_R$ (days of rabbit raising) and $W_R$ (weight of rabbit), a rabbit weight estimation dataset was constructed with seven indexes mentioned above as inputs, and the manually weighted $W_R$ as the output.

The dataset was standardized first. Figure 13 shows the corresponding relationship between input indexes and rabbit weight.

According to Figure 13, among the input indexes, skeleton length and curvature do not have an obvious correlation relationship with rabbit weight, while relative area had a higher correlation with weight. Thus, the data interval of relative area was standardized to [0.6, 1]; the data interval of body width, body length, and contour perimeter was standardized to [0.4, 0.8]; the days of rabbit raising were standardized to [0, 0.4]; skeleton length and curvature data interval were standardized to [0.2, 0.6]. The data interval of weight was standardized to [0.6, 1].

The standardized dataset is divided into training set and verification set at 8:2 for the training and evaluation of meat rabbit weight estimation network.

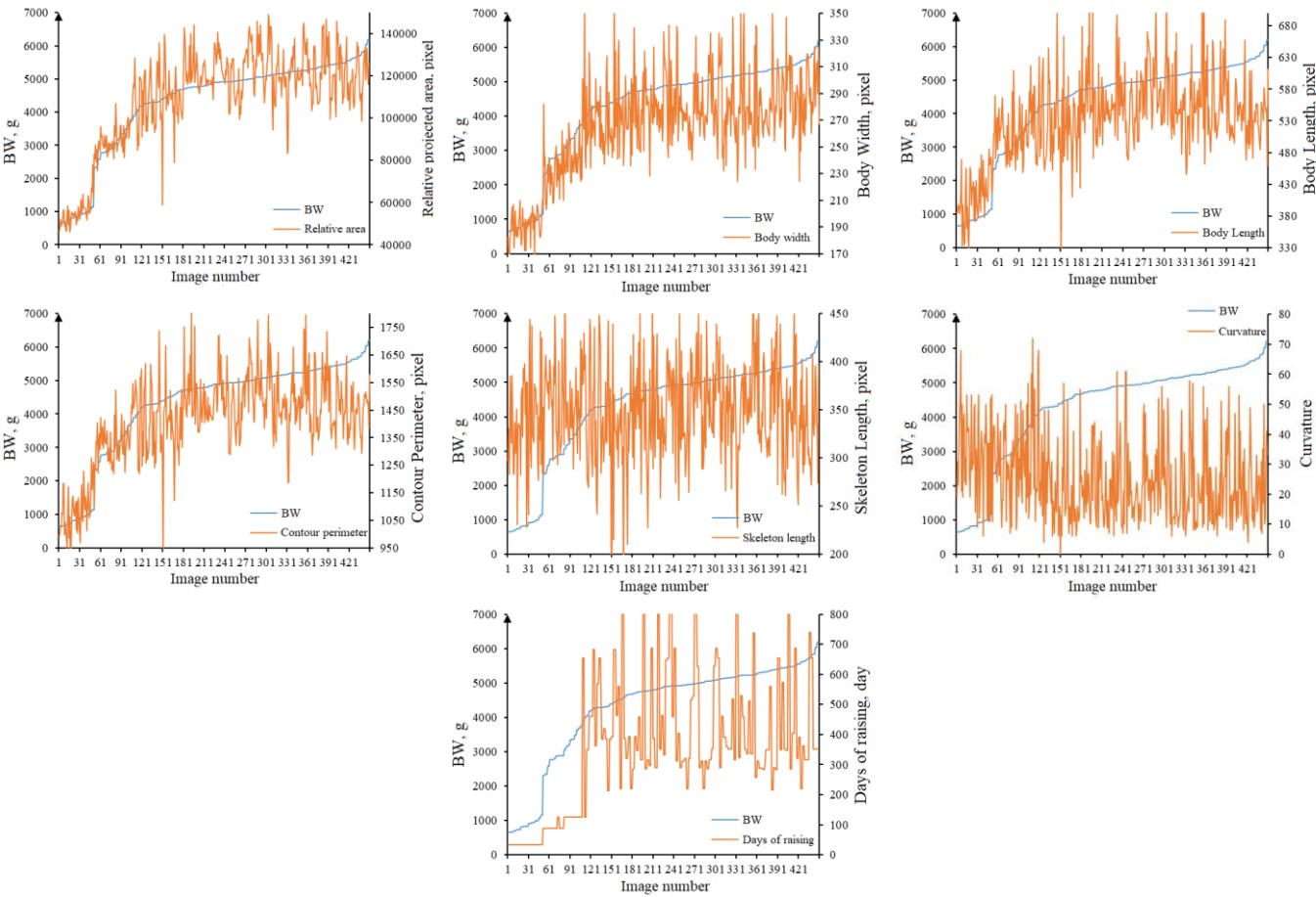

**Figure 13.** The corresponding relationship between input indexes and rabbit weight.

### 3.2.2. Weight Estimation Network Performance Evaluation Indexes

Regression coefficient ($r$), root mean square error (RMSE), mean absolute error (MAE), and relative error $\delta$ were used as evaluation indexes for the performance of the rabbit weight estimation model, among which relative error refers to the proportion of the difference between the predicted weight and the actual weight.

### 3.2.3. Parameter Tuning of BPNN, SVR, and SSA

BPNNs with different parameters were constructed and trained to obtain the optimal parameters of the model. The number of the hidden layer of BPNN was either one or two, and the number of hidden layer nodes were from four to six. The training algorithms selected were Trainlm, Trainbr, and Trainscg, while the adaptive learning function and the transfer function were selected as Learngdm and Tansig based on experience. The maximum iteration times were set at 1000 and the learning rate at 0.01. The performance of BPNNs with different parameters is shown in Table 2.

According to Table 2, when the number of hidden layers is 2, and the number of hidden layer nodes is [5, 6], and the training algorithm used is Trainbr, the performance of BPNN is optimal, with the best RMSE and MAE.

The radial basis function (RBF) was chosen as the kernel function of SVR algorithm, and the penalty parameter C was properly selected by using the grid search regression optimization method. The penalty coefficient C and the kernel function feature value g were selected to be 1024 and 4.8, respectively, and the value of epsilon was set to 0.1 by default. The parameters of SSA were determined according to the amount of data in this research. The population size of SSA algorithm was set to 10 according to experience, the

maximum iteration times was set to 80, the forewarning value ST was set to 0.8, and the ratio of discoverer and joiner was 2:8.

**Table 2.** The performance of BPNNs with different parameters.

| Structure of Hidden Layer | Training Algorithms | r | RMSE | MAE |
|---|---|---|---|---|
| (4) | Trainlm | 0.969 | 356.8 | 278.2 |
| | Trainbr | 0.971 | 347.5 | 276.1 |
| | Trainscg | 0.960 | 407.5 | 320.7 |
| (5) | Trainlm | 0.968 | 358.9 | 278.2 |
| | Trainbr | 0.973 | 334.5 | 261.2 |
| | Trainscg | 0.954 | 436.1 | 332.5 |
| (6) | Trainlm | 0.971 | 341.8 | 267.9 |
| | Trainbr | 0.973 | 335.2 | 261.8 |
| | Trainscg | 0.965 | 379.6 | 302.4 |
| (4,6) | Trainlm | 0.967 | 365.7 | 288.6 |
| | Trainbr | 0.975 | 323.1 | 246.9 |
| | Trainscg | 0.953 | 440.9 | 350.2 |
| (5,6) | Trainlm | 0.971 | 346.9 | 259.5 |
| | Trainbr | 0.980 | 291.2 | 223.0 |
| | Trainscg | 0.955 | 430.3 | 331.7 |
| (6,5) | Trainlm | 0.971 | 356.0 | 271.6 |
| | Trainbr | 0.964 | 385.0 | 275.8 |
| | Trainscg | 0.965 | 381.7 | 301.6 |

### 3.2.4. Performance of the Meat Rabbit Weight Estimation Network

The weight estimation network dataset was used to compare the performance of BPNN, SVR, and SVR-SSA-BPNN, and the results are shown in Table 3.

**Table 3.** Performance of three rabbit weight estimation model.

| Model | r | RMSE | MAE | $\delta$, 100% |
|---|---|---|---|---|
| BPNN | 0.980 | 291.2 | 223.0 | 6.1 |
| SVR | 0.961 | 355.0 | 278.4 | 8.7 |
| SVR-SSA-BPNN | 0.987 | 227.3 | 172.7 | 4.3 |

The results in Table 2 show that the regression performance of the SVR model is the worst among the three models; meanwhile, the prediction performance of the optimized SVR-SSA-BPNN is higher than that of BPNN and SVR, and the RMSE value reaches 227.3 and the MAE value reaches 172.7 with a relative error rate of 4.3%, which indicates that the network stability and accuracy have been improved by network weighting and weight optimization.

To reduce the estimation error of BW for each rabbit in different postures, three images of each rabbit were taken continuously in the image acquisition process. The models were used to estimate the BW of all rabbits, and the mean value of the three predicted weights was taken as the final BW prediction result. The relationship between the real and predicted values of the models was shown in Figure 14.

As shown in Figure 14, the predicted values of the three models are significantly correlated with the true values, but the predicted value of SVR-SSA-BPNN is closer to the true values. With the continuous increase in rabbit BW, the fitting capabilities of the models all decreased and were lower than the true values. This may be due to the fact that for rabbits with larger BW, the mask is affected by continuous opening operations while removing ears and legs, thereby leading to more parts being wrongly removed, which in turn makes the predicted BW biased low.

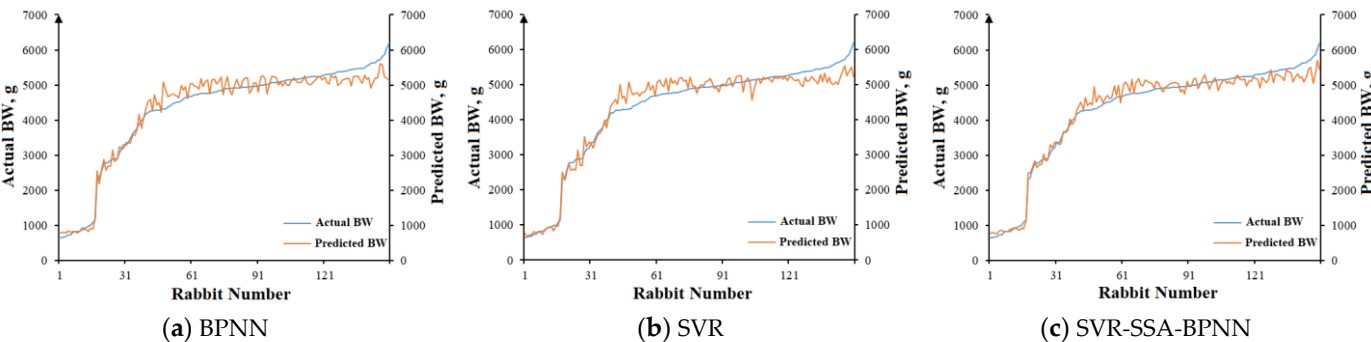

**Figure 14.** The relationship between the real and predicted values of the models.

## 4. Discussion

This study, based on an existing study, took commercial caged rabbits as the targets of the weight estimation task and developed an image automatic collection patrol platform to collect information on automation in commercial scenarios, and provide a feasible technological means for non-contact rabbit weight estimation. Based on Mask RCNN, the effects of different optimization methods on the segmentation effects of caged rabbits were explored, and the mechanism of the method's effect on image segmentation performance was analyzed. The optimal image segmentation network was found under the scenario of rabbits' caged breeding. The results show that the Mask RCNN optimized with PointRend, and attention mechanism optimization has the best image segmentation performance, with an accuracy of 99.1% for object classification and 98.7% for pixel-wise classification, and an inference time of 0.13 s for a single image. The optimized model can meet the needs of actual breeding. Seven image feature indexes related to rabbit weight were further proposed, and the correlation between features was analyzed. The results show that the relative projection area has the most significant correlation with rabbit weight. BPNN is optimized from the perspectives of network weighting and weight optimization, and a rabbit weight regression model based on SVR-SSA-BPNN is proposed. The results show that the regression coefficient of the rabbit weight regression model is 0.987, RMSE is 227.3, MAE is 172.7, and the relative error of the predicted value to the true value is 4.3%. The above results provide a feasible method for the large-scale monitoring of rabbit weight in commercial scenarios.

## 5. Conclusions

The main objective of this study is to develop a weight estimation system suitable for commercial meat rabbit farming. To achieve this, we developed an image automatic collection robot as the hardware component of the weight estimation system, and developed a machine vision-based weight estimation model as the software component of the system. The system achieved the expected goals during the experiment, but there are still some issues. First, due to the shape of the cage, the camera must be far away from the center of the robot to capture images that meet the requirements, which seriously affects the center of gravity of the robot and is the direct cause of the robot's path fluctuations. In the future, we will develop a track-type image collection device, but the cost will be higher than that of the robot. On the other hand, as meat rabbits have gregarious characteristics, we found that when extracting images of multiple rabbits, it is almost impossible to segment complete rabbit contours due to occlusion; so, the method proposed in this paper can only be applied to the case of one rabbit per cage. Fortunately, meat rabbits are raised individually except during their infancy. Finally, edge computing should be integrated into the weight estimation system to reduce ineffective data transmission. The use of image estimation to estimate the weight of rabbits has good application prospects in China. In the future, the way images are collected should be optimized to improve the performance of

image segmentation and to obtain more accurate image features to improve the accuracy of weight estimation.

**Author Contributions:** Conceptualization, software, validation, investigation, formal analysis, data curation, writing—original draft preparation, review and editing, visualization, E.D. and H.H.; methodology, E.D. and S.Z.; resources, supervision, H.W.; funding acquisition, Z.B. All authors have read and agreed to the published version of the manuscript.

**Funding:** This work was funded by the Agriculture Science and Technology Independent Innovation Project of Jiangsu Province (Grant No. CX(22)1008), the China Agriculture Research System of MOF and MARA (CARS-43-D-3).

**Institutional Review Board Statement:** The animal study was reviewed and approved by the Institutional Animal Care and Use Committee of Jiangsu Academy of Agricultural Sciences.

**Informed Consent Statement:** Not applicable.

**Data Availability Statement:** Not applicable.

**Conflicts of Interest:** The authors declare no conflict of interest.

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
