# Peer review of "Estimating Body Weight in Captive Rabbits Based on Improved Mask RCNN"

_agriculture, doi:10.3390/agriculture13040791_

Round 1

Reviewer 1 Report

In this study, the authors suggested a body weight estimation system to solve the problem of weight estimation in the commercial cage breeding of meat rabbits. The method comprises a meat rabbit image acquisition robot and an estimation model. I have the following suggestion for improving this manuscript:

1- Line 18: Please state the six morphological parameters.

2- In the introduction section, the motives are clear, but the contributions should be shown more clearly. In addition, please add more related studies and the novelty of your research compared with the previous studies, not only the importance of the study.

3- Please add more detail for the section of the Algorithm platform (Line 3019), not only the device and program used.

4. The discussion section needs substantial improvement as it does not mention any comparison of the current study with any previous studies.

5. I think that the authors should mention a section for conclusions that explains the outputs and recommendations of the study, the limitations of the study, and whether there is a need for further studies in the future or whether this proposed product is applicable.

Reviewer 2 Report

- The motivation need to better articulated from what is mentioned in the abstract.

- mean average precision it usually abbreviate mAP in the literature. Please correct.

- Line 57, what is artificial weighing.

- Line 61, grammatical error, "an meat".

- Line 63 you should not write this study, which refers to the current manuscript. 

- Lines 67-69 can't be understood due to poor english. 

- Figure 2 is not really needed for the purpose of this paper. 

- Line 115, what is "{#8}" this is not propoer way of citing. 

- What is the accuracy, precision, and recall values? Also, if relevant, then please include the precision-recall curves.

- Include the bland-altman plots for weight agreement.

- What is the number and location of detection heads in the RCNN network. 

- Figure 6 is not need nore it is the contribution of this paper. 

- Table 1, why the comma in the headers.

- Similar studies utilizing faster R-CNN can be cited so that to established the trustworthiness of the models and can provide reliabilit to baseline settings, see Detection of K-complexes in EEG waveform images using faster R-CNN and deep transfer learning. BMC Med InformDecis Mak 22, 297 (2022). https://doi.org/10.1186/s12911-022-02042-x

- The table of abbreviations is missing but required by the journal template.

Round 2

Reviewer 1 Report

The authors responded to all my suggestions and the manuscript has improved significantly.

Reviewer 2 Report

The authors addressed my comments.